# Characterization of a C-Type Lectin Domain-Containing Protein with Antibacterial Activity from Pacific Abalone (*Haliotis discus hannai*)

**DOI:** 10.3390/ijms23020698

**Published:** 2022-01-09

**Authors:** Mi-Jin Choi, Yeo Reum Kim, Nam Gyu Park, Cheorl-Ho Kim, Young Dae Oh, Han Kyu Lim, Jong-Myoung Kim

**Affiliations:** 1Department of Fisheries Biology, Pukyong National University, Busan 48513, Korea; coawls@pukyong.ac.kr (M.-J.C.); dufmadl7@pukyong.ac.kr (Y.R.K.); 2Department of Biological Technology, Pukyong National University, Busan 48513, Korea; ngpark@pknu.ac.kr; 3Department of Biological Sciences, Sungkyunkwan University, Suwon 16419, Korea; chkimbio@skku.edu; 4Department of Marine and Fisheries Resources, Mokpo National University, Muan 58554, Korea; ohyd55@gmail.com

**Keywords:** antimicrobial activity, bacterial binding assay, C-type lectin, Pacific abalone, perlucin

## Abstract

Genes that influence the growth of Pacific abalone (*Haliotis discus hannai*) may improve the productivity of the aquaculture industry. Previous research demonstrated that the differential expression of a gene encoding a C-type lectin domain-containing protein (CTLD) was associated with a faster growth in Pacific abalone. We analyzed this gene and identified an open reading frame that consisted of 145 amino acids. The sequence showed a significant homology to other genes that encode CTLDs in the genus *Haliotis*. Expression profiling analysis at different developmental stages and from various tissues showed that the gene was first expressed at approximately 50 days after fertilization (shell length of 2.47 ± 0.13 mm). In adult Pacific abalone, the gene was strongly expressed in the epipodium, gill, and mantle. Recombinant Pacific abalone CTLD purified from *Escherichia coli* exhibited antimicrobial activity against several Gram-positive bacteria (*Bacillus subtilis*, *Streptococcus iniae*, and *Lactococcus garvieae*) and Gram-negative bacteria (*Vibrio alginolyticus* and *Vibrio harveyi*). We also performed bacterial agglutination assays in the presence of Ca^2+^, as well as bacterial binding assays in the presence of the detergent dodecyl maltoside. Incubation with *E*. *coli* and *B*. *subtilis* cells suggested that the CTLD stimulated Ca^2+^-dependent bacterial agglutination. Our results suggest that this novel Pacific abalone CTLD is important for the pathogen recognition in the gastropod host defense mechanism.

## 1. Introduction

The Pacific abalone *Haliotis discus hannai* is one of the most commercially valuable shellfish species in northeast Asia [1]. Various methods have been implemented to improve its productivity and foster its beneficial traits; these methods include selective breeding, hybridization, and transcriptomic approaches [2,3,4,5,6]. Researchers have also screened for faster growth in Pacific abalone to reduce the time necessary for the shellfish to reach a marketable size. In a previous study, we screened differentially expressed genes associated with faster growth in Pacific abalone and identified three genes involved in immune-related responses [5,6].

The immune system plays a critical role in its protection against pathogens. Most organisms have two types of protection systems: an innate immunity, which provides a general response to pathogens, and an adaptive immunity, which elicits a more specialized response. Mollusks do not have an adaptive immune system; they rely on the innate immune system for their protection from invading pathogens [7]. In mollusks, defense responses are activated when the non-self patterns found on microbes (e.g., fungi and viruses) are recognized [8]. These pathogen-associated molecular patterns (PAMPs) include lipopolysaccharides, peptidoglycans, and mannans that are conserved in Gram-negative and Gram-positive pathogens and fungi. Previous research has indicated that these carbohydrate PAMPs are recognized by a class of proteins that includes the C-type lectin superfamily proteins, which are widely distributed among invertebrates, vertebrates, and plants [9,10].

C-type lectins and C-type lectin domain-containing proteins (CTLDs) in invertebrates recognize PAMPs in invading pathogens; upon pathogen recognition, C-type lectins and CTLDs stimulate innate immune system responses that include agglutination, opsonization, and antibacterial activity [11,12]. CTLDs generally have at least one carbohydrate-recognition domain that consists of approximately 110–130 amino acid residues [13]. The carbohydrate-recognition domains often have two or three conserved disulfide bonds that stabilize a double-loop structure [14]. CTLDs from various invertebrates (e.g., crustaceans, shellfish, and gastropods) reportedly have PAMP-binding characteristics and antimicrobial activity. Recombinant C-type lectins from several crustaceans function as antimicrobial proteins by binding to invading microbes and stimulating agglutination [15,16,17]. In addition, a recombinant CTLD from bay scallophas shown a high binding affinity for lipopolysaccharides; this affinity was significantly decreased upon the addition of mannose and galactose, suggesting that the CTLD mediates the opsonization of invading microbes [18].

In this study, we characterized a gene encoding a CTLD from Pacific abalone (*AbCTLD*) and investigated its possible functions. We deduced the full-length amino acid sequence from the amplified genomic sequence; the cDNA was used to predict the protein structure and create a phylogenetic tree. Expression profiling analysis showed that this gene was initially expressed 50 days after fertilization, mainly in tissues exposed to the aquatic environment. In addition, we tested the antibacterial activity of recombinant AbCTLD (rAbCTLD) produced in *Escherichia coli* against potentially pathogenic aquatic microbes in vitro.

## 2. Results

### 2.1. Sequence Analysis of the Gene Encoding AbCTLD

Previous research demonstrated that the differential expression of the gene encoding AbCTLD was associated with a faster growth in Pacific abalone [5]. The structure of this gene was determined by comparing DNA sequences that had been amplified from genomic DNA and cDNA templates. The *AbCTLD* nucleotide sequence has been deposited in the GenBank database (accession number OK414015). The *AbCTLD* gene has three exons with lengths of 118 bp, 93 bp, and 224 bp (Figure 1). A polyadenylation signal (aataaa) and two introns (474 bp and 606 bp long) were also identified with GT/AG splice sequences at the exon–intron junctions.

The sequence of the gene encoding AbCTLD has an open reading frame of 435 bp, which encodes a protein consisting of 145 amino acids. The translated sequence has a theoretical molecular weight of 17.3 kDa and an isoelectric point of 8.44. No signal peptide sequence or propeptide cleavage sites were identified. The amino acid sequence of AbCTLD was 64.9% and 52.5%, which is similar to perlucins from *Haliotis laevigata* (UniProt P82596) and *Haliotis discus discus* (GenBank ABO26594.1), respectively; it was only 39.2% similar to a CTLD from *Poecilia reticulata* (Guppy, UniProt A0A3P9N573). A multiple sequence alignment showed that six of the seven cysteine residues from AbCTLD were conserved in other proteins, such as perlucins and CTLDs from other mollusks and teleosts (Figure 2A). QPD and WND carbohydrate-recognition-domain motifs were also identified. The predicted carbohydrate-recognition domain (SMART accession number SM00034) covered amino acid residues 14 to 142 with an E-value of 2.77 × 10^–27^.

A phylogenetic tree was constructed using 21 CTLD/perlucin protein sequences from mollusks, teleosts, and mice (Figure 2B). AbCTLD clustered with perlucin and CTLDs from mollusks, whereas the homologous CTLD sequences from teleosts clustered into a different clade. Phylogenetic data showed that AbCTLD clustered with perlucins from *Lingula unguis*, *H. laevigata*, and *H. diversicolor*. Collectin and a CTLD from mice, which are only distantly related to perlucins and CTLDs from mollusks, were used as an outgroup to construct the phylogenetic tree.

### 2.2. Structure Modeling and Ligand Prediction

Protein structure modeling based on the structure of *Homo sapiens* C-type mannose receptor 2 (PDB c5ao6A, UniProt Q9UBG0) predicted that AbCTLD has two alpha helices, α_1_ (FAEASAYCCY) and α_2_ (KDEDDFLRSY; Figure 3A). The modeling suggested that six of the seven AbCTLD cysteine residues would form three disulfide bridges (Cys_14_–Cys_25_, Cys_42_–Cys_141_, Cys_114_–Cys_131_) to stabilize the protein structure. There was a predicted binding site for Ca^2+^ at residues Asp_80_, Asn_107_, Glu_112_, and His_113_ (template 1k9jA, C-score 0.15). There was a predicted site for mannose interaction at residues Glu_112_, Ala_129_, Asp_128_, Asn_127_, Asp_106_, and Gln_104_ (template 3pakA, C-score 0.44; Figure 3B).

### 2.3. Developmental and Tissue-Specific Expression Pattern of AbCTLD

To analyze the expression of *AbCTLD* across the Pacific abalone’s development, samples were collected from early developmental stages including unfertilized eggs, the four-cell stage, and the morula, trochophore, veliger, and post-larval stages (5 days post-fertilization (dpf)); samples were also collected from spat at 50, 100, and 150 dpf. A reverse transcription (RT)-polymerase chain reaction (PCR) analysis was performed on cDNA synthesized from identical quantities of the total RNA extracted from larvae (eggs to postlarvae) and the whole juvenile abalone spat (at 50, 100 and 150 dpf), using primers corresponding to the *AbCTLD* gene. The results indicated that *AbCTLD* transcripts were first expressed approximately 50 dpf and substantially expressed in spat at 100 and 150 dpf (Figure 4A). An analysis of the tissue-specific mRNA expression in adult abalone by quantitative real-time (qRT)-PCR showed that *AbCTLD* transcripts were highly expressed in the epipodium, gill, and mantle (Figure 4B).

### 2.4. Purification of His_6_-Tagged rAbCTLD

rAbCTLD with a C-terminal His_6_ tag was induced using isopropyl β-D-1-thiogalactopyranoside (IPTG) and expressed in *E. coli* BL21 cells (Figure 5). A slow induction of the protein over 3 h with 0.1 mM IPTG was optimal; a small increase in the protein induction occurred with the supplementation of an additional 1 mM IPTG. The predicted molecular weight of the induced protein was 18.3 kDa. There were an additional eight amino acids at the C-terminus, including two amino acids (Leu and Glu) that had been added during the inclusion of an *XhoI* restriction site and His_6_-tag. rAbCTLD was purified from *E. coli* lysates using affinity chromatography with an immobilized metal ion, then harvested with 500 mM imidazole. The purified rAbCTLD was dialyzed in 100 mM Tris-HCl (pH 8.0) and further purified by reversed-phase high-performance liquid chromatography (RP-HPLC; Figure 6). Fractions corresponding to the peaks collected at 43.0–43.5 min showed proteins of the expected size (approximately 18.3 kDa) by SDS-PAGE as well as by mass spectrometry. While the result suggests AbCTLD has a monomeric structure, a possible involvement of the dimeric structure was not completely excluded, as a peak corresponding to its molecular weight was also detected by a mass spectrometry. These fractions were first assayed for antibacterial activity against *Bacillus subtilis*.

### 2.5. Antibacterial Activity of AbCTLD

The antibacterial activity of rAbCTLD was measured using an ultrasensitive radial diffusion assay. The antibacterial activities of 15 µg, 5 µg, and 1 µg of rAbCTLD were tested against Gram-positive (*B. subtilis*, *Streptococcus iniae*, and *Lactococcus garvieae*) and Gram-negative (*Vibrio alginolyticus* and *Vibrio harveyi*) bacteria (Figure 7). The antibacterial activity against *E. coli* was also found to be similar to that of other Gram-negative bacteria (data not shown). The greatest antibacterial activity was exhibited by 15 μg of rAbCTLD (inhibition zone diameter 7.97 ± 0.009 mm). This level of antibacterial activity was similar to the activity of 1 μg of ampicillin, which had a greater antibacterial efficacy in terms of its unit weight (Figure 7B). However, the molecular weight of rAbCTLD was 46-fold greater than the molecular weight of ampicillin (371.4). Therefore, rAbCTLD had a three-fold greater antibacterial efficacy per molecule than ampicillin.

### 2.6. Bacterial Agglutination Stimulated by rAbCTLD

The purified rAbCTLD was tested to determine whether it could stimulate the bacterial agglutination of *E. coli* (BL21) cells expressing green fluorescent protein (GFP) or *B. subtilis* cells labeled with 4′,6-diamidino-2-phenylindole (DAPI). The bacterial cells were incubated with various concentrations of Ca^2+^ up to 10 mM. No bacterial agglutination was observed in the reaction mixtures containing 10 mM Ca^2+^ plus Tris-buffered saline (TBS) or 10 mM Ca^2+^ plus bovine serum albumin (BSA), in the absence of rAbCTLD (Figure 8A,B). In contrast, a bacterial agglutination of both *B. subtilis* and *E. coli* was observed in the mixtures containing 10 mM Ca^2+^ in the presence of rAbCTLD (Figure 8F). Some bacterial agglutination was observed in the mixtures containing 1 mM Ca^2+^ in the presence of rAbCTLD, although the size of the cell clump was smaller than that of the clump formed in the presence of 10 mM Ca^2+^. No distinguishable agglutination was observed in the mixtures containing 0.1 mM Ca^2+^ in the presence of rAbCTLD (Figure 8D). These results clearly indicate that rAbCTLD stimulated bacterial agglutination in a Ca^2+^-concentration-dependent manner.

We tested whether rAbCTLD could bind to bacterial cells using a centrifugation assay in which the supernatant containing the unbound protein was separated from the pellet containing bacteria bound to rAbCTLD. Although most of rAbCTLD was present in the fraction containing precipitated bacterial cells after centrifugation, the purified rAbCTLD tended to self-precipitate under our experimental conditions (data not shown). Therefore, to confirm the specific binding of the recombinant protein to the bacterial cells using an assay based on centrifugation, we inhibited the rAbCTLD self-precipitation by using dodecyl maltoside, which is widely utilized for membrane protein solubilization. We found that rAbCTLD was detected in the supernatant but did not precipitate, indicating that dodecyl maltoside adequately solubilized the purified rAbCTLD in the absence of bacteria (Figure 9). In contrast, most of the rAbCTLD was detected in the pellet containing bacterial cells upon the addition of either Gram-negative *E. coli* or Gram-positive *B. subtilis*. These results show that rAbCTLD bound to bacterial cells.

## 3. Discussion

The C-type lectins are a superfamily of proteins that are involved in regulating a diverse range of physiological functions, including innate and adaptive antimicrobial immune responses [19]. The superfamily of proteins that contain at least one C-type lectin domain is large and diverse; these proteins recognize a broad range of ligands including PAMPs [20]. A differentially expressed gene that influenced growth in Pacific abalone exhibited sequence similarity to the C-type lectin domains at residues 14 to 142 [5]. Seven cysteine residues were identified in the deduced amino acid sequence, including six highly conserved cysteine residues that may form disulfide bonds to stabilize the CTLD structure [21,22]. In addition, the presence of two alpha helices was deduced from three-dimensional protein structure modeling. Therefore, the protein was classified as a Pacific abalone CTLD.

The sugar-binding sites of CTLDs from various invertebrates vary in sequence. Although the QPD and EPN motifs that bind galactose and mannose are frequently present in CTLDs, variations in other sequence motifs (e.g., EPD, EPQ, QPT, and QPG) are common [16,23,24,25,26,27]. WND is another key sugar-binding motif that occurs in various invertebrate CTLDs [28]. In AbCTLD from *H. discus hannai*, the putative sugar-binding motifs were QPD (amino acids 104–106) and WND (amino acids 126–128), which are conserved in perlucins and CTLDs from other abalone and teleosts. These motifs may mediate the binding of AbCTLD to carbohydrate moieties on the surfaces of invading microbes. The Asn_127_ residue in the WND motif may function as a Ca^2+^-binding site. Furthermore, the AbCTLD amino acid sequence was similar (64.9%) to the amino acid sequence of perlucin from *H. laevigata* (UniProt P82596), which contains QPD and WND motifs. This perlucin exhibited binding affinities with both galactose and mannose [29].

The high mortality rate of abalone spat is a major factor that limits the productivity of the abalone aquaculture industry [30]. Various approaches have been used to address this problem, including the selection of faster-growing strains with increased immunity. Genes involved in immunoprotection are important for host cells’ protection from environmental stress and pathogen infection. Among these, the gene encoding AbCTLD was associated with a faster growth and antimicrobial activity. The expression of AbCTLD was particularly high in the mantle, which connects the inner surface of the shell with the visceral mass, and the epipodium, which is located on the dorsal foot. These tissues are frequently exposed to the aquatic environment. Therefore, AbCTLD may be important for their protection against pathogens. During the early developmental egg to post-larval (5 dpf) stage, *AbCTLD* expression was low. The highest levels of *AbCTLD* gene expression occurred at 100 to 150 dpf, indicating that the CTLD may provide protection from pathogens as early as 100 dpf. Time course studies in several species have shown that *CTLD* expression is upregulated in various tissues in response to PAMPs or injection with bacteria [31,32,33,34]. In addition, the expression of *CTLDs* in the gill and mantle of manila clam was significantly upregulated in response to lipopolysaccharides or the injection of *Vibrio tapetis* bacteria [35,36]. Therefore, the high level of expression of *CTLDs* observed in diverse species, from invertebrates to mammals, may be involved in the Ca^2+^-dependent innate immunity against invading pathogens.

rAbCTLD exhibits antimicrobial activity against pathogens (e.g., *S. iniae*, *L. garvieae*, *V. alginolyticus*, and *V. harveyi*) that cause serious diseases in aquatic animals [37,38,39,40]. Although they exhibited low sequence similarities to AbCTLD, other abalone *CTLDs* demonstrated increasing expression in response to their infection with *Vibrio*; moreover, recombinant CTLDs mediated the pathogen agglutination by binding to mannose [41,42,43]. In this study, we identified a novel gene encoding a CTLD in *H. discus hannai* (AbCTLD) and analyzed its expression pattern in various tissues and across developmental stages.

We found that rAbCTLD had an antibacterial activity, stimulated bacterial agglutination, and functioned in the innate immunity by binding to carbohydrate moieties. CTLDs functioned in innate immune responses by acting as pattern recognition receptors, binding to the carbohydrate moieties of glycoproteins on microbes, and inducing the bacterial agglutination in a Ca^2+^-dependent manner [44,45,46]. *E**. coli* expression studies showed that recombinant CTLDs from mollusks, such as *H. discus hannai* [42,43], scallop [12], and razor clam [47], can stimulate bacterial agglutination. Our rAbCTLD stimulated the bacterial agglutination in a Ca^2+^-dependent manner; this may enable Pacific abalone to eliminate invading pathogens. We used a ligand prediction algorithm to identify four residues (Asp_80_, Asn_107_, Glu_112_, and His_113_) that may form a Ca^2+^ binding site, which could subsequently influence the carbohydrate-binding activity in AbCTLD.

## 4. Materials and Methods

### 4.1. Cloning the AbCTLD Gene

We extracted genomic DNA (gDNA) from 100 mg of Pacific abalone mantle tissue using the procedure established by Asahida et al. [48]. The *AbCTLD* gene was amplified using primers listed in Table 1. PCR products were extracted using a gel extraction kit and ligated into the pTOP TA V2 vector (Enzynomics, Daejeon, Korea). We transformed *E. coli* DH5α cells and extracted the plasmid using the method established by Sambrook and Russell [49]. Purified TA plasmid was sequenced using the M13F-20 and M13R universal primers.

### 4.2. Phylogenetic Tree and Bioinformatic Analysis

The *AbCTLD* gene sequence was obtained by amplification using both gDNA and cDNA templates. The deduced amino acid sequence was used for phylogenetic tree construction along with amino acid sequences from other CTLDs, perlucins, and collectins that were in the GenBank and UniProt databases (Appendix A). A phylogenetic tree was constructed using the neighbor-joining method; pairwise distances were estimated using the Jones–Taylor–Thornton matrix model. All positions containing gaps and missing data were eliminated by pairwise deletion with 1000 replications of bootstrap testing using Molecular Evolutionary Genetics Analysis software (version 5, Tokyo, Japan) [50].

The translated amino acid sequence of AbCTLD and the related sequences obtained from UniProt and the National Center for Biotechnology Information database were aligned by BioEdit Sequence Alignment Editor and Vector NTI suite 8 (INFORMAX, Bethesda, MD, USA). These sequences were compared with the sequences of HlPer (*H. laevigata* perlucin, P82596), HddPer1 (*H. discus discus* perlucin 1, ABO26590.1), PrCTLD (*P. reticulata* (Guppy) C-type lectin domain-containing protein, A0A3P9N573), NbCTLD (*Neolamprologus brichardi* (Fairy cichlid) C-type lectin domain-containing protein, A0A3Q4MIS0), MgCTL2 (*Mytilus galloprovincialis* C-type lectin 2, AJQ21493.1), and CgPer (*Crassostrea gigas* (Pacific oyster]) perlucin, K1QRE6). A three-dimensional protein model of AbCTLD was constructed using Vector NTI Suite 8 and Phyre2 (http://www.sbg.bio.ic.ac.uk/phyre2, 30 September 2019); the theoretical molecular weight and isoelectronic point of the amino acid sequence were calculated using the Compute pI/Mw tool (https://web.expasy.org/compute_pi/, 30 September 2019) [51]. The SMART tool (http://smart.embl-heidelberg.de, 1 October 2019) and COACH tool (https://zhanglab.ccmb.med.umich.edu/COACH/, 2 October 2019) software applications were used to predict domains and ligand binding sites [52].

### 4.3. Analysis of AbCTLD Gene Expression across Pacific Abalone Development

To analyze the expression profile of the gene encoding AbCTLD, abalone samples were collected at 18 °C water temperature from early developmental stages including unfertilized eggs, the four-cell stage, the morula, trochophore, veliger, and post-larval stage (5 dpf) at the Institute of Ocean and Fisheries Technology (Jeollanam-do, Republic of Korea), together with juvenile abalone collected at 50, 100, and 150 dpf (shell lengths of 2.47 ± 0.13 mm, 8.14 ± 1.87 mm, and 13.28 ± 3.31 mm, respectively). Whole abalone tissue, except for muscle, was used for RNA extraction and cDNA synthesis in accordance with a previously described protocol [53]. *Ribosomal protein L3* (*RPL3*) was used as a reference gene, as in our previous study [5]. All primers used in this study are listed in Table 1. RT-PCR consisted of an initial denaturation step at 95 °C for 3 min, followed by 30 cycles of 95 °C denaturation for 30 s, 60 °C annealing for 30 s, and 72 °C amplification for 30 s. The final extension step was carried out at 72 °C for 5 min; amplified products were visualized by 1.5% agarose gel electrophoresis.

### 4.4. Analysis of AbCTLD Gene Expression in Adult Abalone Tissue

To analyze the tissue-specific expression pattern of *AbCTLD* mRNA, qRT-PCR analysis was carried out using cDNA from the epipodium, gill, mantle, gonads, hepatopancreas, and hemocytes of adult Pacific abalone (*n* = 5; age = 36 months; shell length = 87.52 ± 2.47 mm; weight = 77.28 ± 4.32 g). The specific primers used in qRT-PCR were designed using *RPL3* (reference gene, GenBank KP698943.1). Primer efficiencies for qRT-PCR were tested using the standard curve method (qRPL3-F/R 136 bp, r^2^ = 0.9996, E = 96.14%; qAbCTLD-F/R 133 bp, r^2^ = 0.9956, E = 99.53%). The PCR conditions consisted of a preheating step at 50 °C for 2 min, an activation step at 95 °C for 10 min, and 40 cycles of amplification as follows: 95 °C for 15 s, 60 °C for 30 s, and 72 °C for 30 s. We used a PikoReal™ Real-Time PCR system (Thermo Scientific, Waltham, MA, USA) and SYBR green-based Maxima SYBR Green/ROX quantitative PCR Master Mix (Thermo Scientific). The relative expression of each gene was calculated using the 2^–∆∆CT^ formula [54]. Statistical analysis of qRT-PCR data was performed with one-way analysis of variance and Duncan’s multiple range test using SPSS software (ver. 18; IBM-SPSS, Chicago, IL, USA); *p*-values < 0.05 were considered statistically significant [55].

### 4.5. Purification of AbCTLD

The full length *AbCTLD* gene was amplified from a cDNA template that had been synthesized from ganglion total RNA, with additional restriction sites for *NdeI* and *XhoI*. The purified PCR product was first cloned into the TA vector (pTOP TA V2; Enzynomics), then subcloned into the pET-44a^(+)^ expression vector (Novagen, Madison, WI, USA). The recombinant vector containing a full-length *AbCTLD* gene with a C-terminal His_6_-tag was transformed into *E. coli* BL21 codon plus RP cells (Stratagene, La Jolla, CA, USA). rAbCTLD expression was induced using various concentrations (0.1–1 mM) of IPTG at 20 °C for 4 h in lysogeny broth, supplemented with chloramphenicol (34 µg/mL) and ampicillin (100 µg/mL). Proteins were separated on a 12% polyacrylamide gel, then transferred to a membrane for immunoblotting; a His_6_-tag monoclonal antibody (Invitrogen, Carlsbad, CA, USA) and anti-mouse IgG-horseradish peroxidase (Bio-Rad, Hercules, CA, USA) were used to detect His_6_-tagged rAbCTLD. Immunoblotting results were visualized using an ImageQuant LAS 500 system (GE Healthcare, Buckinghamshire, UK).

For recombinant protein purification, transformed *E. coli* BL21 cells were treated with 0.1 mM IPTG at 20 °C for 3 h, then harvested by centrifugation at 10,000× *g* for 10 min. The cells were lysed by freezing, thawing, and sonication (2 min, 20% amplitude for 2 s) in 1× lysis-equilibration-wash buffer (50 mM NaH_2_PO_4_, 300 mM NaCl, pH 8.0) containing 8 M urea. Next, 10 mM CaCl_2_ and 2 mM dithiothreitol were added; the lysate was incubated at room temperature for 30 min, then centrifuged at 10,000× *g* for 30 min. The supernatant was diluted by adding three volumes of 1× lysis-equilibration-wash buffer, then subjected to affinity purification. The rAbCTLD protein fused to a His_6_-tag was purified with 2 M urea on a Protino^®^ Ni-TED nickel column (MACHEREY-NAGEL, Düren, Germany) under denaturing conditions in accordance with the manufacturer’s instructions. Recombinant proteins were eluted using lysis-equilibration-wash buffer containing 500 mM imidazole and 2 M urea. Stepwise dialysis was performed against lysis-equilibration-wash buffer containing 1 M and 0.5 M urea in the presence of 100 mM Tris (pH 8.0) and using a 3.5 kDa MWCO dialysis membrane (Spectra/Por^®^ membrane; Thermo Scientific). Purified proteins were separated using RP-HPLC with a linear gradient of 5–80% acetonitrile and 0.1% trifluoroacetic acid on a Capcell Pak C18 column (Shiseido, Tokyo, Japan). Samples were eluted at a flow rate of 1 mL/min and monitored using an ultraviolet light detector at 220 nm, 254 nm, and 280 nm. The collected fractions were lyophilized and dissolved in 100 mM Tris (pH 8.0).

### 4.6. Antibacterial Assays

The antibacterial activity of rAbCTLD purified by RP-HPLC was tested using the following bacterial strains: *B. subtilis* (KCTC1021), *S. iniae* (BS9), *L. garvieae* (ATCC 43921), *V. alginolyticus* (KCTC 2928), and *V. harveyi* (ATCC 14126). Antibacterial activity was measured using an ultrasensitive radial diffusion assay [56]. Bacterial strains were cultured in tryptic soy broth (TSB) and lysogeny broth (for *S. iniae*) at 37 °C with shaking (150 rpm). Bacterial culture adjusted to 0.06 of OD_600_ (approximately 84% transmittance) was plated with 9.5 mL of radial diffusion assay buffer containing agarose. Identical volumes (5 µL) of ampicillin and 100 mM Tris (pH 8.0) were used as positive and negative controls, respectively.

### 4.7. Bacterial Agglutination Assays

Purified rAbCTLD was tested to determine whether it could stimulate the agglutination of *E**. coli* BL21–GFP (Gram-negative) or *B**. subtilis* (Gram-positive) cells. Bacterial cells were harvested from growth culture and resuspended in TBS at an OD_600_ of 0.6. Equal volumes of bacterial resuspension cultures were mixed with purified rAbCTLD and incubated at 28 °C for 1 h in the presence of final concentrations of 10 mM, 1 mM, 0.1 mM, or 0 mM Ca^2+^. An identical quantity of bovine serum albumin was used as a negative control. Fluorescence microscopy was used to visualize the *B**. subtilis* cells after they had been stained with DAPI (10 μg/mL) for 3 min at room temperature in the dark.

### 4.8. Bacterial Binding Assays

Binding of rAbCTLD to Gram-negative (*E**. coli* BL21–GFP) and Gram-positive (*B**. subtilis*) bacteria was tested by solubilizing rAbCTLD in dodecyl maltoside. In total, 0.2 μg of rAbCTLD was solubilized in 0.05% dodecyl maltoside and mixed with a bacterial suspension culture in TSB at an OD_600_ of 0.6. The solubilized rAbCTLD was mixed with 360 μL of bacterial culture and incubated at room temperature for 30 min with gentle shaking. TBS was used as a negative control. Samples were centrifuged at 3000× *g* for 1 min. Immunoblotting with an anti-His tag antibody was used to analyze each supernatant and pellet fraction.

## 5. Conclusions

We identified *AbCTLD*, a novel C-type lectin gene in Pacific abalone. The predicted protein sequence contained QPD and WND motifs that have important roles in carbohydrate recognition, together with six cysteine residues that are conserved in C-type lectins found in other mollusks and in fish. Ligand prediction analyses also identified Ca^2+^ and mannose binding sites. The *AbCTLD* gene was first expressed approximately 50 days after fertilization; in adult abalone, it was strongly expressed in tissues that are exposed to the external environment, such as the epipodium, gill, and mantle. We expressed His_6_-tagged rAbCTLD in *E. coli* and showed that the recombinant protein exhibited antibacterial properties and stimulated bacterial agglutination. Therefore, AbCTLD had an important role in the invertebrate immune response by recognizing carbohydrate moieties on the surfaces of pathogens; it stimulated bacterial agglutination in a Ca^2+^-dependent manner.

## Figures and Tables

**Figure 1 ijms-23-00698-f001:**
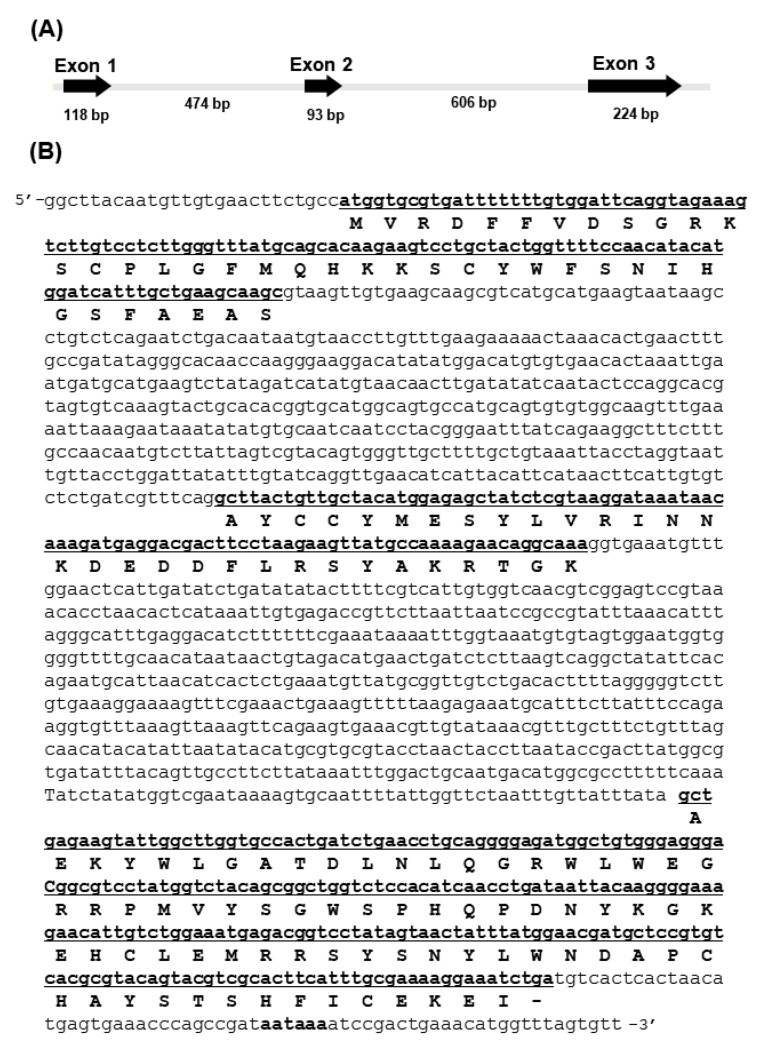
Structure and sequence of a gene encoding a C-type lectin domain-containing protein (CTLD) from Pacific abalone (*AbCTLD*). (**A**) The gene encoding AbCTLD consists of three exons (underlined) with lengths of 118 bp, 93 bp, and 224 bp. (**B**) The coding regions, corresponding amino acids, and polyadenylation signal (aataaa) are shown in bold.

**Figure 2 ijms-23-00698-f002:**
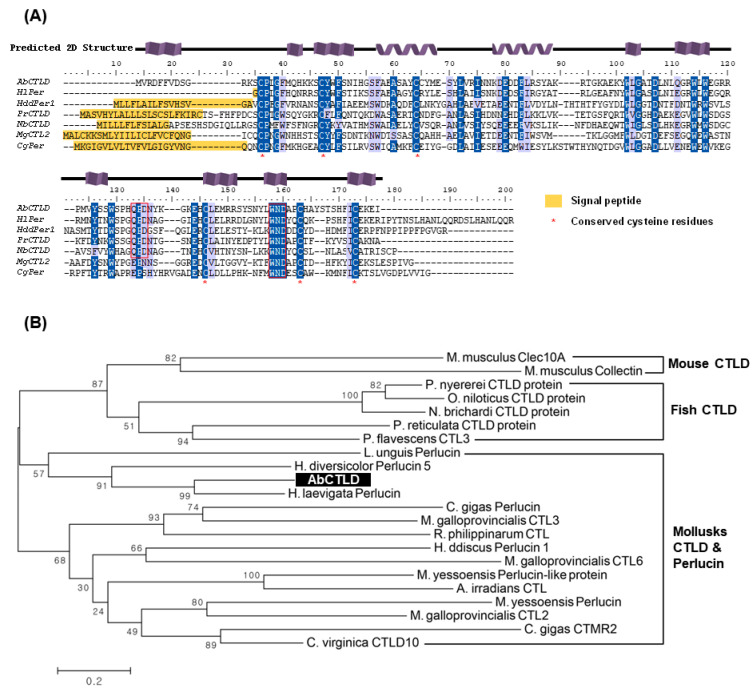
Analysis of the AbCTLD amino acid sequence. (**A**) Multiple alignment of the amino acid sequences of Pacific abalone C-type lectin (AbCTLD) and CTLDs in other mollusks and fish. Particular characteristics of the C-type lectin domains, including two predicted alpha helices and six conserved cysteine residues, are indicated. The aligned amino acid sequences are in greenlip abalone *Haliotis laevigata* perlucin (HlPer, P82596), guppy *Poecilia reticulata* (PrCTLD, A0A3P9N573), Pacific oyster *Crossostrea gigas* perlucin (CgPer, K1QRE6), Fairy cichlid *Neolamprologus brichardi* CTLD (NbCTLD, A0A3Q4MIS0), *Haliotis discus discus* perlucin 1 (HddPer1, ABO26590.1), and Mediterranean mussel *Mytilus galloprovincialis* CTL2 (MgCTL2, AJQ21493.1). Conserved amino acids are highlighted. (**B**) Construction of a neighbor-joining tree, based on the amino acid sequences of CTLDs from mollusks and teleosts. Mouse CTLD sequences were included as outgroups; bootstrap values are indicated for each node.

**Figure 3 ijms-23-00698-f003:**
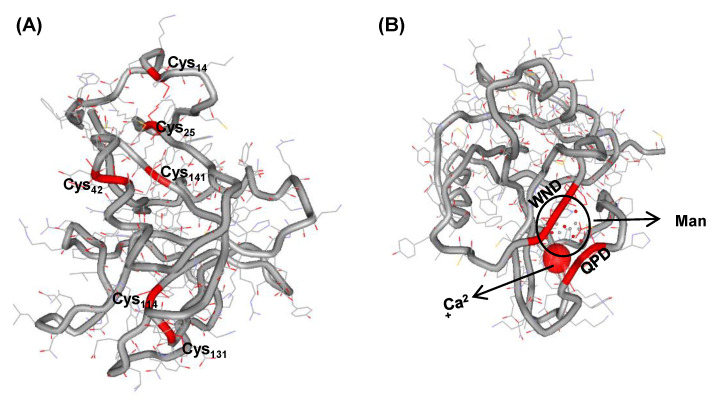
Three-dimensional structure of the AbCTLD protein. (**A**) Six conserved cysteine residues are shown in the three-dimensional structure of AbCTLD. (**B**) A ligand prediction algorithm was used to identify the binding sites of Ca^2+^ (C-score: 0.15) and Man (mannose, C-score: 0.44).

**Figure 4 ijms-23-00698-f004:**
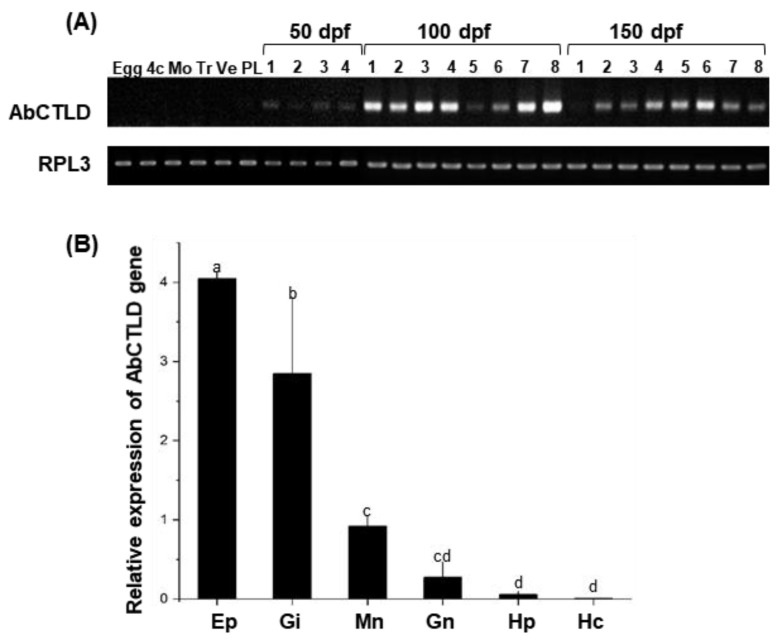
*AbCTLD* expression analysis of distinct developmental stages and tissues in Pacific abalone. (**A**) Reverse transcription (RT)-polymerase chain reaction (PCR)-based detection of *AbCTLD* expression throughout abalone development. The early stages of development included unfertilized eggs (Egg), the four-cell stage (4c), and the morula (Mo), trochophore (Tr), veliger (Ve), and post-larval stages (PL, meaning 5 days post-fertilization (dpf)), as well as juvenile abalone spat at 50 (*n* = 4), 100 (*n* = 8), and 150 (*n* = 8) dpf. (**B**) Tissue-specific abundances of AbCTLD mRNA transcripts in Pacific abalone were also analyzed by quantitative real-time-PCR using cDNA from total RNA samples that had been isolated from various tissues (e.g., epipodium (Ep), gill (Gi), mantle (Mn), gonads (Gn), hepatopancreas (Hp), and hepatocytes (Hc)) in 2-year-old Pacific abalone (*n* = 5). Bars and error bars represent the means and standard deviations of triplicate measurements. Different letters above each bar indicate statistically significant difference (*p* < 0.05)

**Figure 5 ijms-23-00698-f005:**
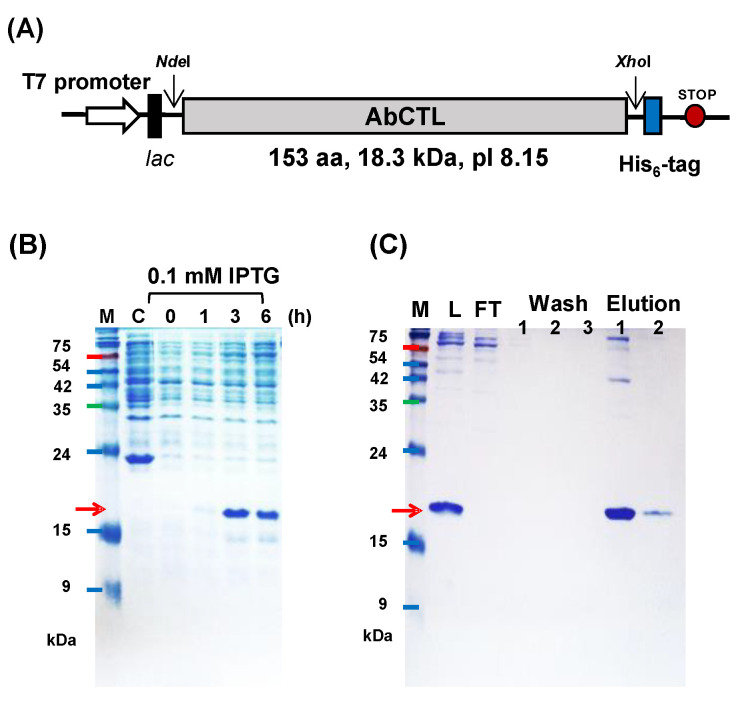
Purification of recombinant AbCTLD (rAbCTLD) expressed in *Escherichia coli.* (**A**) A schematic representation of the pET–AbCTLD expression vector derived from pET44a^(+)^. The cloning sites, promoter, and His_6_-tag fusion site are indicated. (**B**) Sodium dodecyl sulfate-polyacrylamide gel electrophoresis (12% acrylamide gel) of *E*. *coli* BL21 cell lysates after induction with 0.1 mM isopropyl β-D-1-thiogalactopyranoside (IPTG) at 20 °C for 0, 1, 3, and 6 h. Lane C shows lysate from BL21 cells as a negative control. Lane M contains molecular weight markers; the arrow indicates the expected size of rAbCTLD. (**C**) rAbCTLD was purified on a nickel column under denaturing conditions with 2 M urea. Cell lysate (L), flow through (FT), and wash fractions were loaded onto a 12% acrylamide gel; rAbCTLD was identified in eluted fractions 1 and 2.

**Figure 6 ijms-23-00698-f006:**
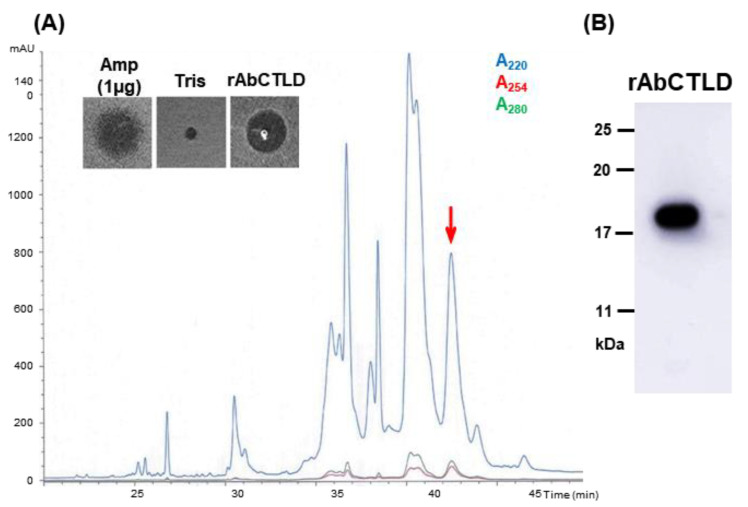
Purification of rAbCTLD using reversed-phase high-performance liquid chromatography (RP-HPLC). (**A**) Eluted fractions were evaluated using an ultraviolet light detector at 220 nm, 254 nm, and 280 nm. The fraction indicated by an arrow was lyophilized and dissolved in 100 mM Tris (pH 8.0). The antibacterial activity of this fraction against *Bacillus subtilis* is shown in the figure inset. (**B**) Immunoblotting analysis of purified rAbCTLD detected using anti-His_6_ tag antibodies.

**Figure 7 ijms-23-00698-f007:**
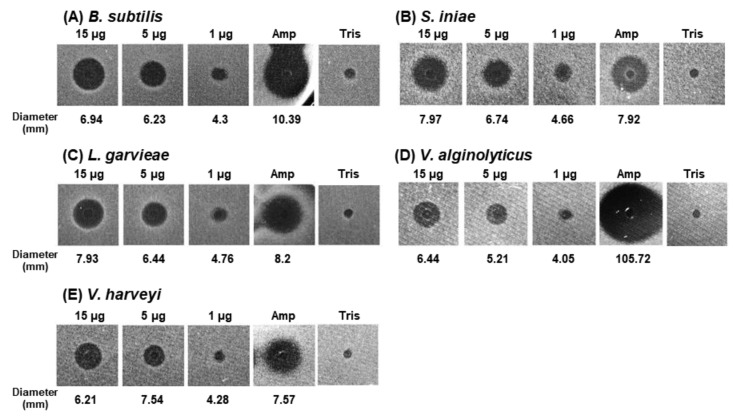
Antibacterial activity of purified rAbCTLD measured using an ultrasensitive radial diffusion assay. The antibacterial activity of 1 µg to 15 µg of purified rAbCTLD was measured using three Gram-positive strains—*B*. *subtilis* (**A**); *Streptococcus iniae* (**B**); and *Lactococcus garvieae* (**C**)—and two Gram-negative strains—*Vibrio alginolyticus* (**D**) and *Vibrio harveyi* (**E**). Equal volumes of solutions containing 100 mM Tris but without rAbCTLD, and 1 µg of ampicillin were used as negative and positive controls, respectively.

**Figure 8 ijms-23-00698-f008:**
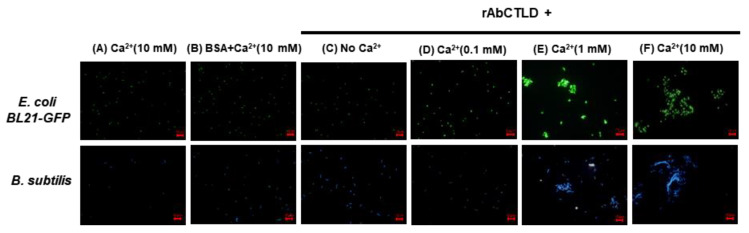
Fluorescence microscopy shows that bacterial agglutination was stimulated by rAbCTLD in both *E*. *coli* BL21 and *B*. *subtilis* cells in a Ca^2+^-dependent manner. Bacterial agglutination tests were performed on *E*. *coli* BL21–GFP (upper panel) and *B*. *subtilis* (lower panel) cells incubated with or without purified rAbCTLD (0.18 μg) in the presence of various concentrations of Ca^2+^. Bacterial cells were incubated with 10 mM Ca^2+^ (**A**), or bovine serum albumin (BSA) plus 10 mM Ca^2+^ (**B**) in the absence of rAbCTLD. Bacterial cells were incubated with 0 mM (**C**), 0.1 mM (**D**), 1 mM (**E**), or 10 mM (**F**) of Ca^2+^ in the presence of rAbCTLD. (scale bar: 10 μm).

**Figure 9 ijms-23-00698-f009:**
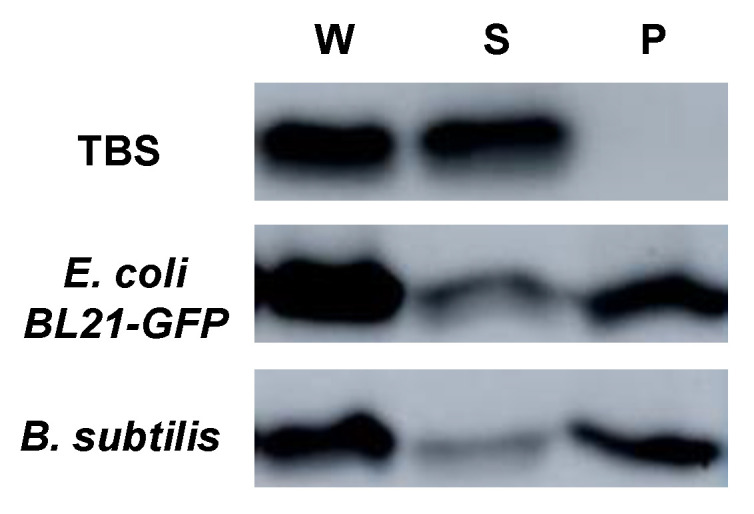
Bacterial binding analysis of rAbCTLD resuspended in 0.05% dodecyl maltoside. Bacterial binding analysis was performed on purified rAbCTLD together with *E*. *coli* or *B*. *subtilis*. Precipitates were separated by centrifugation, and purified recombinant proteins were dissolved in 0.05% dodecyl maltoside, then incubated with *E*. *coli* or *B*. *subtilis* as indicated. Tris-buffered saline (TBS) replaced bacterial cells in the control samples (TBS). The levels of rAbCTLD in the whole reaction mixture (W), supernatant (S), and pellet (P) were compared by immunoblotting analysis using an anti-His tag antibody.

**Table 1 ijms-23-00698-t001:** List of primers used to identify and analyze AbCTLD.

Purpose	Primer	Sequence (5′-3′)
Identification of *AbCTLD* gene structure in genomic DNA	5UTR-F	GGC TTA CAA TGT TGT GAA CTT CTG
	Par-R2	GTA CGC GTG ACA CGG AGC AT
	Par-F2	AGC GCT TAC TGT TGC TAC ATG G
	3UTR-R	CAA CAC TAA ACC ATG TTT CAG TCG G
Cloning of full-length *AbCTLD* into pET expression vector	FullF-NdeI	CAT ATG GTG CGT GAT TTT TTT GTG GAT TC
	FullR-XhoI	CTC GAG GAT TTC CTT TTC GCA AAT GAA GTG
RT-PCR of *AbCTLD*	AbCTLD-F	CCT CTT GGG TTT ATG CAG CAC
	AbCTLD-R	CGG ACT GTC TCA TTT CCA GAC
RT-PCR of *RPL3* (Housekeeping gene)	RPL3-F	TGT CAC CAT CCT TGA GGC AC
	RPL3-R	CAG GAA CAG GCT TCT CCA GG
qRT-PCR of *AbCTLD*	qAbCTLD-F	GGT GCC ACT GAT CTG AAC CT
	qAbCTLD-R	AGG ACC GTC TCA TTT CCA GA
qRT-PCR of *RPL3* (Housekeeping gene)	qRPL3-F	AGT CCT TCC CTA AGG ATG ACA AG
	qRPL3-R	GCC TCC ACA ACT TCC TTC TTA TT

## Data Availability

Data supporting reported results can be provided by J.-M.K. upon request.

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
