# Peer review of "Characterization of a C-Type Lectin Domain-Containing Protein with Antibacterial Activity from Pacific Abalone (Haliotis discus hannai)"

_ijms, 2022, doi:10.3390/ijms23020698_

Round 1

Reviewer 1 Report

The investigation of novel AbCTLD in Hdiscus hannai is the good paper but some points should mention clearer as mentioned below.

  1. Line 169, the authors mentioned about the antibacterial activity against Bacillus subtilis after purification of recombinant AbCTLD (rAbCTLD). However, the result in the figure 6 did not show the effect of bacterial killing in rAbCTLD. In addition, the pictures of inhibition zone in ampicillin treatment also showed different compared with figure 7 despite the same concentration of ampicillin (1µg Amp) (Figure 6 and 7). The authors should check results again.
  2. In the experiment of antibacterial activity, the authors test in both Gram-positive strains ( subtilis, Streptococcus iniae, and Lactococcus garvieae) and Gram-negative strains (Vibrio alginolyticus and Vibrio harveyi). However, the antibacterial activity in E. coli that was also used in the bacterial agglutination (figure 8) did not include in this experiment. Did the authors do the experiment of antimicrobial activity in E. coli and how were the results when compared with Amp or Tris?
  3. In figure 4A, the authors did not label in each lane among 50, 100 and 150 dpf because there are 4 lanes in 50 dpf, 8 lanes in 100 and 150 dpf. The readers might confuse this result. The author should explain more detail in the picture or in the text.
  4. Line 404, the authors would like to explain the method of antibacterial assays. However, the authors did not mention how long incubated with shaking before testing or which OD600 was selected to do experiment.
  5. There is no primer list used in the experiment or table 1 in this manuscript.

Minor points

  1. In figure 8, the authors did not mention how many micrograms of rAbCTLD were used in the experiment of bacterial agglutination.
  2. Line 141 in the sentence of “…total RNA extracted from whole spat or at least 50 days after fertilization??? eggs or larvae…”, the authors might lose some words as mention in red color.
  3. In the abbreviation of “AbCTLD”, the authors sometimes wrote “AbCTL” that need to check to confirm.
  4. In the table S1, the authors did not have the detail of accession in the table or what is meaning of “2” in definition of C. gigas Please check in again.

Author Response

Dear Reviewer, 

We would like to express our sincere gratitude for the helpful comments on our manuscript entitled, “Characterization of a c-type lectin domain-containing protein with antibacterial activity from Pacific abalone (Haliotis discus hannai) : ijms-148530”.

We have revised the manuscript according to the comments. Please refer to the changes incorporated in the following pages and an attached file with highlighted changes.  If you have any question, please let me know.

Sincerely

Jong-Myoung Kim, PhD

Professor, Dept. of Marine BioMaterials & Aquaculture

PuKyong National University

e-mail : [email protected]

- Comments from Reviewer 1

The investigation of novel AbCTLD in Hdiscus hannai is the good paper but some points should mention clearer as mentioned below.

Query 1) Line 169, the authors mentioned about the antibacterial activity against Bacillus subtilis after purification of recombinant AbCTLD (rAbCTLD). However, the result in the figure 6 did not show the effect of bacterial killing in rAbCTLD. In addition, the pictures of inhibition zone in ampicillin treatment also showed different compared with figure 7 despite the same concentration of ampicillin (1µg Amp) (Figure 6 and 7). The authors should check results again.

Response 1-1) We apologize for our mistake in labeling the samples. We corrected the misinformation in a new Figure 6.

Query 2) In the experiment of antibacterial activity, the authors test in both Gram-positive strains ( subtilisStreptococcus iniae, and Lactococcus garvieae) and Gram-negative strains (Vibrio alginolyticus and Vibrio harveyi). However, the antibacterial activity in Ecoli that was also used in the bacterial agglutination (figure 8) did not include in this experiment. Did the authors do the experiment of antimicrobial activity in Ecoli and how were the results when compared with Amp or Tris?

 Response 1-2) Antimicrobial activity was also tested against E. coli using the same URDA method but the result was not included in Figure 7 as it was similar to those of other Gram negative bacteria. In Figure 8, E. coli expressing GFP was used for detecting an aggregation activity by measuring a fluorescence. Please refer to the changes included in the revision as follows: .

: Lines 189~ 192 : The antibacterial activities of 15 µg, 5 µg, and 1 µg of rAbCTLD were tested against Gram-positive (B. subtilis, Streptococcus iniae, and Lactococcus garviea) and Gram-negative (Vibrio alginolyticus and Vibrio harveyi) bacteria (Fig. 7). The antibacterial activity against E. coli was also found to be similar to that of other Gram- negative bacteria (data not shown). The greatest antibacterial activity was exhibited by 15 mg of rAbCTLD (inhibition zone diameter: 7.97± 0.009 mm).

 Query 3) In figure 4A, the authors did not label in each lane among 50, 100 and 150 dpf because there are 4 lanes in 50 dpf, 8 lanes in 100 and 150 dpf. The readers might confuse this result. The author should explain more detail in the picture or in the text.

Response 1-3) We apologize for an unclear explanation in Figure 4AThe figure was changed with lane labeling together with more detailed information in the caption as follows:    

: Figure 4. AbCTLD expression analysis of distinct developmental stages and tissues in Pacific abalone.(A) Reverse transcription (RT)-polymerase chain reaction (PCR)-based detection of AbCTLD expression throughout abalone development. The early stages of development included unfertilized eggs (Egg), the four-cell stage (4c), and the morula (Mo), trochophore (Tr), veliger (Ve), and post-larval stages (PL, 5 days post-fertilization [dpf]), as well as juvenile abalone spat at 50 (n = 4), 100 (n = 8), and 150 (n = 8) dpf.

Query 1-4) Line 404, the authors would like to explain the method of antibacterial assays. However, the authors did not mention how long incubated with shaking before testing or which OD600 was selected to do experiment.

Response 1-4) Thanks for your kind suggestion. More detailed information was included in the revision as follows:  

: Bacterial strains were cultured in tryptic soy broth and lysogeny broth (for S. iniae) at 37°C with shaking (150 rpm). Bacterial culture adjusted to 0.06 of OD600 (approximately 84% transmittance) was plated with 9.5 ml radial diffusion assay buffer containing agar. Identical volumes (5 µL) of ampicillin and 100 mM Tris (pH 8.0) were used as positive and negative controls, respectively.

Query 1-5) There is no primer list used in the experiment or table 1 in this manuscript.

Response 1-5) Sorry for the missing information. Table 1 was included in the revision

Minor points

  1. In figure 8, the authors did not mention how many micrograms of rAbCTLD were used in the experiment of bacterial agglutination.

Response) Figure 8. Fluorescence microscopy shows that bacterial agglutination was stimulated by rAbCTLD in both E. coli BL21 and B. subtilis cells in a Ca2+-dependent manner. Bacterial agglutination tests were performed with E. coli BL21–GFP (upper panel) and B. subtilis (lower panel) cells incubated with or without purified rAbCTLD (0.18 mg) in the presence of various concentrations of Ca2+. Bacterial cells were incubated with 10 mM Ca2+ (A) or bovine serum albumin (BSA) plus 10 mM Ca2+ (B) in the absence of rAbCTLD. Bacterial cells were incubated with 0 mM (C), 0.1 mM (D), 1 mM (E), or 10 mM (F) Ca2+ in the presence of rAbCTLD.

  1. Line 141 in the sentence of “…total RNA extracted from whole spat or at least 50 days after fertilization??? eggs or larvae…”, the authors might lose some words as mention in red color.

Response) Thanks for the comments on the missing information. Please refer to the information included in the revision.

: Reverse transcription (RT)-polymerase chain reaction (PCR) analysis was performed with cDNA synthesized from identical quantities of total RNA extracted from at least 50 larvae (eggs to postlarvae) and the whole juvenile abalone spats (at 50, 100 and 150 dpf), using primers corresponding to the AbCTLD gene.

2. In the abbreviation of “AbCTLD”, the authors sometimes wrote “AbCTL” that need to check to confirm.

Response) Thanks for the correction. All “AbCTL” were changed to “AbCTLD”. 

3. In the table S1, the authors did not have the detail of accession in the table or what is meaning of “2” in definition of Cgigas Please check in again.

Response) New table S1 was included.

Reviewer 2 Report

In the paper “Characterization of a C-type lectin domain-containing protein with antibacterial activity from Pacific abalone (Haliotis discus hannai)” the Authors continue previous studies that identified differentially expressed genes associated with faster growth in Pacific abalone.

In particular, in this manuscript they analyse one such genes encoding a C-type lectin domain-containing protein (namedAbCTLD) that shows an ORF corresponding to a polypeptide of 145 amino acids showing significant homology to genes encoding CTLD-containing proteins in other molluscs and fish. They investigate the expression of this protein at different developmental stages and in various tissues and purify it after cloning in E. coli. The recombinant protein is used to test its antimicrobial activity against several marine microorganisms that cause serious diseases in aquatic animals and its capacity to stimulate bacterial agglutination.

The paper is in general well written and easy to read and the results are presented in a logical sequence. However, there are some points that need to be clarified:

Fig. 1: the signal peptide cleavage site is marked with a yellow slash that is difficult to see. In the figure legend it is not specified what the red boxes indicate.

At variance with other homologous proteins that contain six Cys residues forming three disulphide bonds, AbCTLD has seven cysteine residues one of which remains unpaired. Can the Authors exclude that a dimer is formed via an intermolecular disulphide bridge? 

The mass of the RP-HPLC purified recombinant protein (peak in Fig. 6) should be verified by mass spectrometry, and not only estimated by SDS-PAGE.

In the insert in Fig. 6A are the pictures identified by rAbCTLD and Tris inverted? Otherwise the insert would indicate that TRIS alone is more active than the recombinant protein. In addition, it is not indicated what the amount (in micrograms) of rAbCTLD added is, nor the diameter of the region in which bacterial growth is inhibited by ampicillin and rAbCTLD.

Fig. 7, line 202, are the bacterial cells resuspended in TRIS or is TRIS without AbCTLD used as a negative control? The Authors should report the diameter of the inhibition zones. 

Fig. 8, it would be more logical to exchange panels F and D so that the figure starts with the controls followed by 0, 0.1, 1 and 10 mM Ca2+.

Paragraph 2.6. Bacterial agglutination stimulated by rAbCTLD.

In the text the Authors correctly report that in the absence of Ca2+ or in the presence of 0.1 mM Ca2+ the recombinant protein did not agglutinate the bacteria tested. Then they report that agglutination was observed in the presence of 10 mMCa2+ and some bacterial agglutination in the mixture containing 1 mM Ca2+, suggesting that in the presence of this lower concentration of Ca2+ the protein is less effective in promoting agglutination. Looking at Fig. 9, it is difficult to establishthat AbCTLD agglutinates bacteria better at 10 mM Ca2+ compared to 1 mM. It would be clearer if the agglutinationcould be quantified.  

Legend to Fig. 9, line 242, may be the term resolved should be changed to dissolved. 

Throughout the text, the scientific names of animals and bacteria are not written in italics. Finally, reference n. 9 is not reported in the text.

Author Response

Dear Editor,

Dear Reviewer, 

We would like to express our sincere gratitude for the helpful comments on our manuscript entitled, “Characterization of a c-type lectin domain-containing protein with antibacterial activity from Pacific abalone (Haliotis discus hannai) : ijms-148530”. We have revised the manuscript according to the comments. Please refer to the changes incorporated in the following pages and an attached file with highlighted changes.  If you have any question, please let me know.

Sincerely

Jong-Myoung Kim, PhD

Professor, Dept. of Marine BioMaterials & Aquaculture

PuKyong National University

e-mail : [email protected]

- Comments from Reviewer 2

In the paper “Characterization of a C-type lectin domain-containing protein with antibacterial activity from Pacific abalone (Haliotis discus hannai)” the Authors continue previous studies that identified differentially expressed genes associated with faster growth in Pacific abalone.

In particular, in this manuscript they analyse one such genes encoding a C-type lectin domain-containing protein (namedAbCTLD) that shows an ORF corresponding to a polypeptide of 145 amino acids showing significant homology to genes encoding CTLD-containing proteins in other molluscs and fish. They investigate the expression of this protein at different developmental stages and in various tissues and purify it after cloning in E. coli. The recombinant protein is used to test its antimicrobial activity against several marine microorganisms that cause serious diseases in aquatic animals and its capacity to stimulate bacterial agglutination.

The paper is in general well written and easy to read and the results are presented in a logical sequence. However, there are some points that need to be clarified:

Query 2-1) Fig. 1: the signal peptide cleavage site is marked with a yellow slash that is difficult to see. In the figure legend it is not specified what the red boxes indicate.

Response 2-1) The figure was changed accordingly

Query 2-2) At variance with other homologous proteins that contain six Cys residues forming three disulphide bonds, AbCTLD has seven cysteine residues one of which remains unpaired. Can the Authors exclude that a dimer is formed via an intermolecular disulphide bridge? The mass of the RP-HPLC purified recombinant protein (peak in Fig. 6) should be verified by mass spectrometry, and not only estimated by SDS-PAGE.

Response) As described in lines ~126, Please refer to changes as follows: Fractions corresponding to the peaks collected at 43.0–43.5 min showed proteins of the expected size (approximately 18.3 kDa) by SDS_PAGE as well as by mass spectrometry. While the result suggests a monomeric structure of AbCTLD, possible involvement of a dimeric structure was not completely excluded as a peak corresponding to its molecular weight was also detected by a mass spectrometry. These fractions were first assayed for antibacterial activity against Bacillus subtilis

Query 2-3) In the insert in Fig. 6A are the pictures identified by rAbCTLD and Tris inverted? Otherwise the insert would indicate that TRIS alone is more active than the recombinant protein. In addition, it is not indicated what the amount (in micrograms) of rAbCTLD added is, nor the diameter of the region in which bacterial growth is inhibited by ampicillin and rAbCTLD.

Response 2-3) We apologize for our mistake in labeling the samples in Figure 6A. We corrected the information in a new Figure 6 and added a more detailed information including the diameter of the inhibitory region.

Query 2-4) Fig. 7, line 202, are the bacterial cells resuspended in TRIS or is TRIS without AbCTLD used as a negative control? The Authors should report the diameter of the inhibition zones. 

Response 2-4) Thanks for the comment. The sentence was changed as follows : Equal volumes of solutions containing 100 mM Tris and 1 µg ampicillin were used as negative and positive controls, respectively.

Query 2-4) Fig. 8, it would be more logical to exchange panels F and D so that the figure starts with the controls followed by 0, 0.1, 1 and 10 mM Ca2+. 

Response 2-4) Thanks for the suggestion. Please refer to a new Figure 8 changed according to your suggestion.

Query 2-5) Paragraph 2.6. Bacterial agglutination by rAbCTLD.

In the text the Authors correctly report that in the absence of Ca2+ or in the presence of 0.1 mM Ca2+ the recombinant protein did not agglutinate the bacteria tested. Then they report that agglutination was observed in the presence of 10 mMCa2+ and some bacterial agglutination in the mixture containing 1 mM Ca2+, suggesting that in the presence of this lower concentration of Ca2+ the protein is less effective in promoting agglutination. Looking at Fig. 9, it is difficult to establish that AbCTLD agglutinates bacteria better at 10 mM Ca2+ compared to 1 mM. It would be clearer if the agglutinationcould be quantified.  

Response 2-5)  No bacterial agglutination was observed in the reaction mixtures containing 10 mM of Ca2+ plus Tris-buffered saline (TBS) or 10 mM of Ca2+ plus bovine serum albumin, in the absence of rAbCTLD (Fig. 8A and 8B). In contrast, bacterial agglutination of both B. subtilis and E. coli was observed in the mixtures containing 10 mM Ca2+ in the presence of rAbCTLD (Fig. 8D). Some bacterial agglutination was observed in the mixtures containing 1 mM Ca2+ in the presence of rAbCTLD although size of the cell clump was smaller than that of the formed in the presence of 10 mM Ca2+. No distinguishable agglutination was observed in the mixtures containing 0.1 mM Ca2+ in the presence of rAbCTLD (Fig. 8E and 8F).

Query 2-6) Legend to Fig. 9, line 242, may be the term resolved should be changed to dissolved

Response 2-6) Thanks for the suggestion. The rem was changed to “dissolved”.

Query 2-7) Throughout the text, the scientific names of animals and bacteria are not written in italics. Finally, reference n. 9 is not reported in the text.

Response) Thanks for the comment. Reference9, was cited in the text.